# Salidroside Alleviates Renal Fibrosis in SAMP8 Mice by Inhibiting Ferroptosis

**DOI:** 10.3390/molecules27228039

**Published:** 2022-11-19

**Authors:** Sixia Yang, Tingting Pei, Linshuang Wang, Yi Zeng, Wenxu Li, Shihua Yan, Wei Xiao, Weidong Cheng

**Affiliations:** 1School of Traditional Chinese Medicine, Southern Medical University, Guangzhou 510515, China; 2Institute of Basic Research in Clinical Medicine, China Academy of Chinese Medical Sciences, Beijing 100700, China; 3Key Laboratory of Glucolipid Metabolic Disorder, Ministry of Education, Guangdong Pharmaceutical University, Guangzhou 510006, China

**Keywords:** renal fibrosis, ferroptosis, aging, salidroside, iron transport

## Abstract

Renal fibrosis progression is closely associated with aging, which ultimately leads to renal dysfunction. Salidroside (SAL) is considered to have broad anti-aging effects. However, the roles and mechanisms of SAL in aging-related renal fibrosis remain unclear. The study aimed to evaluate the protective effects and mechanisms of SAL in SAMP8 mice. SAMP8 mice were administered with SAL and Ferrostatin-1 (Fer-1) for 12 weeks. Renal function, renal fibrosis, and ferroptosis in renal tissue were detected. The results showed that elevated blood urea nitrogen (BUN) and serum creatinine (SCr) levels significantly decreased, serum albumin (ALB) levels increased, and mesangial hyperplasia significantly reduced in the SAL group. SAL significantly reduced transforming growth factor-β (TGF-β) and α-smooth muscle actin (α-sma) levels in SAMP8 mice. SAL treatment significantly decreased lipid peroxidation in the kidneys, and regulated iron transport-related proteins and ferroptosis-related proteins. These results suggested that SAL delays renal aging and inhibits aging-related glomerular fibrosis by inhibiting ferroptosis in SAMP8 mice.

## 1. Introduction

The elderly population rate is increasing, owing to improvements in living and medical standards [1]. Many organs change during aging, and the kidneys are particularly affected by aging [2]. Aging causes structural changes in the kidney which lead to conditions such as glomerular and tubular hypertrophy, glomerulosclerosis, and tubulointerstitial fibrosis [3]. Patients with chronic kidney disease (CKD) develop renal anemia and exhibit reduced erythropoietin secretion [4]. Actively combating renal anemia can improve the quality of life and survival rate of patients with CKD [5]. Therefore, research on mechanisms involved in renal aging and alleviation of circulating iron deficiency has gained attention because of their significance in the prevention of aging-related glomerular fibrosis.

Senescence-accelerated mouse (SAM) strains consist of senescence-prone inbred strains (SAMP) and senescence-resistant inbred strains (SAMR) [6]. SAMP8 mice are one of the senescence-accelerated prone mice, based on the grading score data of aging, life span, and pathological phenotype, Kyoto University conducted selective inbreeding on AKR/J mouse strain donated by Jackson Laboratory in 1968 [7], which is characterized by learning and memory deficits and impaired immune function [8]. The serum oxidative stress level of SAMP8 mice was higher than that of SAMR1 mice [9], and the expression levels of muscle atrophy, and inflammation fibrosis-related and aging gene marker genes in elderly SAMP8 mice, were significantly different from those in the young control group [10]. Elderly SAMP8 mice can show adverse cardiac remodeling [11], liver fibrosis [12], and renal fibrosis [13], which are age-related pathologies, and their incidence rate and severity increase with age [6].

Studies have shown that age-related mitochondrial dysfunction is closely related to renal fibrosis [14]. There is dysfunction in the mitochondria of elderly SAMP8 mice [15]. Renal pathological changes of 9-month-old SAMP8 mice include renal tubulointerstitial fibrosis and focal segmental glomerulosclerosis. Renal fibrosis of SAMP8 mice is related to age. The Wnt/β-catenin/RAS signaling pathway was activated in the kidney of SAMP8 mice [16]. Compared with the aging SAMR1 control group, the consumption level of GSH/GSSG ratio and MDA level in the 6-month-old-month old SAMP8 mice were increased [17]. Early CKD was observed in aged SAMP8 mice [18]. A spontaneous model such as SAM has obvious advantages over the gene-modified model, as it can better reflect the changes in age-related diseases. It is hoped that it can be used more widely for the research of biological gerontology resources [6]. There was ferroptosis and increased lipid peroxidation levels in the muscle of SAMP8 mice [19].

A recent study suggested that ferroptosis plays an important role in renal tubular injury in tubular epithelial cells (TECs), and suggested that ferroptosis plays a key role in driving kidney injury and can be used as a treatment strategy [20]. Ferroptosis is an iron-dependent cell death caused by lipid peroxidation that is controlled by integrated oxidant and antioxidant systems [21]. The iron-containing enzyme lipoxygenase is the main ferroptosis promoter by producing lipid hydroperoxides, and its function relies on the Acyl-CoA synthetase long-chain family member 4 (ACSL4)-dependent lipid biosynthesis activation [22]. In contrast, the selenium-containing enzyme Glutathione Peroxidase 4 (GPX4) is currently recognized as a central repressor of ferroptosis, and it is dependent on glutathione, which is produced by the activation of the cystine–glutamate anti-porter. Similar to transferrin (TF), it acts as a positive ferroptosis regulator by increasing iron intake [23]. Ferritin is the major intracellular iron storage protein complex and includes ferritin light polypeptide 1 (FTL1) and ferritin heavy polypeptide 1 (FTH1) [24]. Increased ferritin expression limits ferroptosis. Recent studies have indicated that FTH1 increases autophagy, which can degrade ferritin to elevate iron levels, resulting in oxidative injury through the Fenton reaction [25].

Salidroside (SAL) is a phenylpropanoid glycoside with various pharmacological benefits [26] which acts as an anti-oxidative, anti-fatigue, anti-inflammatory, anti-aging, and anti-diabetic agent. SAL has been found in sweet-scented Osmanthus, Oleifera, Striga, and Rhodiola L [27]. Current evidence suggests that SAL affects renal interstitial fibrosis in unilateral ureteral obstruction model mice [28]. However, there were few reports of SAL involved in renal interstitial fibrosis in the context of aging. In this study, we aimed to investigate the therapeutic effects of SAL in renal interstitial fibrosis in SAMP8 mice and explore its possible mechanism.

## 2. Results

### 2.1. Screened Potential Targets

After removing duplicates, 434 targets of SAL and 2586 targets of aging-related renal fibrosis were collected (Appendix A). Overlap between targets of SAL and aging-related renal, 183 potential therapeutic targets of SAL were selected and presented by Venn diagrams (Figure 1A).

### 2.2. Enrichment Analysis of Potential Targets

The bubble maps provide a graphical representation of the highly enriched terms of function categories. The main molecular functions included amide binding, growth factor-receptor binding, and protein phosphatase binding (Figure 1B). The main biological processes included neutrophil degranulation, neutrophil activation, and response to oxidative stress (Figure 1C). The main cellular components included the glutamatergic synapse, the integral component of the synaptic membrane, and the vesicle lumen (Figure 1D). The main enriched pathways included ferroptosis, aging, apoptosis, dopaminegic synapse, glycolysis/gluconeogenesis, the TNF signaling pathway, MAPK signaling pathway, mTOR signaling pathway, and neurotrophin signaling pathway (Figure 2).

### 2.3. Network Analysis of Potential Targets

The PPI network consisted of 183 nodes and 1332 edges, with an average node degree of 14.7 (Figure 3A). The PPI network contained five clusters, with a local clustering coefficient of 0.466, which was related to tau protein binding metal ion binding, response to oxygen-containing compounds, and regulation of the cell cycle, respectively (Figure 3B). The subnetworks selected by MCODE include module 1 (MCODE score = 27.758) (Figure 3C) and module 2 (MCODE score = 3.667) (Figure 3D). Clusters 1 and 2 were related to the regulation of programmed cell death and abnormal cell death, respectively. Based on the above findings, the following studies were focused on ferroptosis.

### 2.4. Effects of SAL Treatment on Renal Function Parameters from the Serum of SAMP8 Mice

To investigate the protective effect of SAL against aging-related renal damage, biochemical parameters, including BUN and SCr, were evaluated. The serum BUN and SCr levels in the SAMP8 group were significantly higher than in the SAMR1 group (*p* < 0.01; Figure 4A,B). The BUN and SCr levels in the SAL-L, SAL-H, and Fer-1 groups were significantly reduced compared to the SAMP8 group, particularly in the SAL-H group (*p* < 0.01). The serum ALB levels in the SAMP8 group were significantly decreased when compared with the SAMR1 group (*p* < 0.001; Figure 4C). However, the serum ALB levels in the SAL-L, SAL-H, and Fer-1 groups significantly increased when compared with the SAMP8 group, particularly in the SAL-H group (*p* < 0.01). This indicated that senescence-related renal dysfunction had emerged in the SAMP8 group, and SAL had protective effects on the abnormal symptoms.

### 2.5. Effects of SAL Treatment on Histopathological Changes in the Kidneys of SAMP8 Mice

The results of H&E staining revealed that the number of mesangial cells, the volume of glomeruli, and the proliferation of the mesangial matrix were further increased in the SAMP8 group. However, the number of glomerular cells, glomerular volume, and mesangial cell proliferation was significantly reduced in the SAL-L, SAL-H, and Fer-1-treated groups than in the SAMP8 group (Figure 5A). Masson staining also revealed that numerous cyanine-stained collagen fibers and very little red-stained normal kidney tissue appeared in the kidney glomeruli of the SAMP8 group when compared with the SAMR1 group (*p* < 0.01; Figure 5B). This indicated that aging significantly induces glomerular mesangial expansion and mesangial matrix accumulation. However, the SAL-L, SAL-H, and Fer-1 groups showed significantly reduced accumulation of blue-stained collagen fibers in the glomerular basal region than that in the SAMP8 group (*p* < 0.05, *p* < 0.01; Figure 5C). These results suggested that SAL treatment ameliorated renal lesions in the SAMP8 group because of aging.

### 2.6. SAL Ameliorates Renal Interstitial Fibrosis in SAMP8 Mice

We assessed protein expression levels of the profibrotic factor TGF-β1. PCR and Western blot analysis revealed that TGF-β1 expression was significantly increased in the kidney tissue of mice in the SAMP8 group when compared with the SAMR1 group. TGF-β1 protein expression was significantly decreased in the SAL-L, SAL-H, and Fer-1 groups, (Figure 6A,B). Western blot analysis was used to evaluate the expression of α-SMA expression, which is the main molecular marker for myofibroblasts. The results of the Western blot indicated that α-SMA protein expression in the SAMP8 group was significantly increased compared to that in the SAMR1 group, while the level was significantly decreased after being treated with SAL (Figure 6C,D). These results suggested that SAL mediates the secretion of pro-fibrotic factors and activation of myofibroblasts in aging-induced renal tissue, thereby inhibiting aging-induced renal interstitial fibrosis.

### 2.7. Effects of SAL Treatment in SAMP8 Mice on Lipid Peroxidation and GPX4

We assessed oxidative stress and lipid peroxidation in the kidney tissues of the mice. Malondialdehyde (MDA) was enhanced, while superoxide dismutase (SOD) and glutathione (GSH) were decreased in the kidneys of the SAMP8 group when compared with the SAMR1 group (Figure 7). However, the results in the SAL-L, SAL-H, and Fer-1 groups revealed a decrease in MDA, SOD, and GSH increase in the SAMP8 group (Figure 7A–C). Furthermore, we measured GPX4 expression in the mouse kidneys. Immunohistochemistry showed that GPX4 expression decreased in the kidneys of the SAMP8 group; however, the expression of GPX4 was increased in the SAL and Fer-1 groups (Figure 7D,E). These findings further confirmed that lipid peroxidation and ferroptosis are involved in aging-related kidney disease, and that SAL ameliorated aging-related kidney ferroptosis.

### 2.8. SAL Ameliorates Renal Iron Overload in SAMP8 Mice

We measured the iron content in kidney tissue. Iron staining revealed that the iron content in the kidneys of the SAMP8 group was significantly increased (Figure 8A), and the iron content in the renal tubules was relieved after treatment with SAL and Fer-1 (Figure 8B). Western blot showed increased TFR1 expression in the kidneys of the SAMP8 group when compared with the SAMR1 group, while SAL and Fer-1 treatment decreased TFR1. FPN1 and FTH1 expression decreased in the SAMP8 group when compared with the SAMR1 group. SAL and Fer-1 treatment increased the expression of these proteins in the SAMP8 group (Figure 8B,C). These results suggested that aging causes iron overload in renal tubules; however, SAL reduces iron overload in the kidneys of the SAMP8 group.

### 2.9. SAL Ameliorates Renal Ferroptosis in SAMP8 Mice

To determine whether ferroptosis occurs in the kidneys of SAMP8 mice, and whether SAL attenuates aging-induced kidney damage. The expressions of ACSL4, GPX4, and SLC7A11 were detected. In the SAMP8 group, ACSL4 expression was increased (Figure 9A,D,E), and SLC7A11 and GPX4 expression were decreased (Figure 9B–D,F). SLC7A11 and GPX4 expression were significantly improved, and ACSL4 expression was decreased after treatment with SAL and Fer-1 (*p* < 0.05; Figure 9A,D,E).

### 2.10. Molecular Docking Analysis

Molecular docking can mimic the binding ability of different bound compounds and proteins. The classical targets of ferroptosis and SAL were used for molecular docking. The details of these proteins are listed in Table 1. The drug-target binding affinity and the best-scored docked position between this SAL and GPX4, SLC7A11, ACSL4, FTH1, TFR1, and FPN1 are indicated in Figure 10. To some extent, the results supported the reliability of WB and revealed the mode of action of SAL on ferroptosis.

## 3. Discussion

Aging is a significant risk factor for several common human diseases [29]. CKD has become a growing problem worldwide, as it can lead to end-stage renal disease (ESRD) due to renal fibrosis. Aging is an important factor closely related to the occurrence and progression of renal fibrosis [30]. Therefore, finding suitable drugs to prevent the early onset of renal fibrosis because of aging is necessary to reduce the incidence of age-related kidney diseases. Previous studies have found that renal fibrosis occurs in SAMP8 mice [13], and this phenomenon was also found in this study.

Rhodiola Rosea is a well-known herb, and SAL is one of its main active components and is reported to have anti-aging and antioxidant activities [31]. Recent studies have found that SAL inhibits doxorubicin-induced cardiomyopathy by modulating ferroptosis-dependent pathways [32]. In CKD, the major structural kidney lesions include advanced glomerular and interstitial fibrosis, which ultimately leads to renal fibrosis, causing functional failure. It is well-known that aging is closely associated with changes in kidney structure and function [33]. A previous study showed that accumulation of senescent cells is involved in the development of renal fibrosis by upregulating the secretion of pro-inflammatory mediators and pro-fibrotic factors, ultimately preventing cell regeneration [34]. However, the protective effects of SAL against renal fibrosis because of renal aging are unclear. In this study, we found that BUN and SCr significantly decreased, ALB increased, and renal function improved in the SAMP8 group after SAL treatment. Ni et al. stated that SAL can ameliorate renal interstitial fibrosis in mice with diabetic nephropathy [35]. Rui Li et al. found that SAL treatment significantly reduced the release of inflammatory cytokines and inhibited the TLR4/NF-κB and MAPK signaling pathways. The administration of SAL may be a new therapeutic strategy for the treatment of renal fibrosis [36]. This is consistent with our findings.

With the progress of high-throughput technologies, biomedical data have greatly accumulated and continue to grow rapidly. Network pharmacology provides a feasible way to obtain an overall understanding of Traditional Chinese medicine (TCM) prescriptions from these massive clinical and experimental data [37]. Therefore, the combination of network pharmacology and experimental research is a promising approach for identifying potential targets and uncovering therapeutic mechanisms. In this study, 183 potential therapeutic targets of salidroside were screened by network pharmacology for further research. Enrichment analysis and PPI network analysis found that salidroside can regulate metal ion binding, tau protein binding, and oxygen-containing compounds via the ferroptosis pathway. Based on the network pharmacology findings, the following experimental research was focused on ferroptosis.

Mesangial expansion, massive accumulation of the mesangial matrix, glomerular hypertrophy, glomerular fibrosis, and interstitial fibrosis were significantly improved in the SAMP8 group after SAL treatment. These results suggested that intervention can delay the development of a series of structural lesions that occur in the early onset of renal aging in the SAMP8 group. A previous study showed that SAL inhibited the transformation of renal tubular epithelial cells to myofibroblasts in vitro by inhibiting the expression of α-SMA and TGF-β1 based on the dosage. This study revealed that SAL treatment for 12 weeks significantly attenuated mesangial cell proliferation and matrix deposition in the aged SAMP8 group. Furthermore, the TGF-β1 and α-sma expression levels significantly decreased in the SAMP8 group after 12 weeks of SAL administration. These results indicate that SAL treatment has a protective effect against renal fibrosis progression because of aging in the SAMP8 group. In the kidneys of patients with CKD, iron deposits lead to increased iron uptake and/or insufficient iron output. Iron accumulation triggers Fenton-mediated oxidative damage may lead to renal injury, indicating that CKD renal iron accumulation initially induces ferroptosis and that iron plays a deleterious role in CKD progression. Therefore, regulation of the expression of iron metabolism proteins plays a significance in restoring renal iron metabolism and alleviating ferroptosis.

The accumulation of intracellular iron promotes lipid peroxidation, leading to cell death [38]. The results showed that TFR1 significantly increased in the SAMP8 group, whereas FTH1 and FPN1 expression significantly decreased. After 12 weeks of SAL administration, TFR1 levels decreased, and FTH1 and FPN1 levels increased. Kidney iron staining also suggested that iron deposition in the kidneys of the SAMP8 model increased, and iron deposition reduced after SAL intervention, indicating that SAL can ameliorate renal iron overload in an aging model. Ferroptosis is a form of iron-dependent regulated cell death induced by excessive lipid peroxidation, which is morphologically and mechanistically distinct from apoptosis.

In diabetic nephropathy renal biopsies, ferroptosis-related molecules SlC7A11 and GPX4 expression were reduced compared to in non-DN patients [39]. The involvement of ferroptosis has also been demonstrated in an animal model of streptozotocin-induced DN. Significant changes in markers associated with ferroptosis included decreased GPX4 expression levels and increased ACSL4 expression, lipid peroxidation products, and ferroptosis in DN mice [40]. Iron accumulation in multiple organs, including the brain and kidneys, can lead to increased oxidative damage and decreased function in the aging process [41]. Therefore, iron levels can serve as potential ferroptosis biomarkers, and are an important causative factor of age-related diseases. GPX4 utilizes reduced glutathione to convert lipid hydroperoxides to lipid alcohols, thereby reducing lipid peroxidation and inhibiting ferroptosis [42]. Cysteine is obtained by most cells by importing extracellular cystine via the amino acid transporter SLC7A11 [43]. Interventions in the ferroptosis pathway effectively inhibit the progression of these diseases, suggesting that ferroptosis is a potential therapeutic target [44]. As a redox cycle nitrogen oxide, SAL helps reduce oxidative stress and has been reported to protect against neurodegenerative diseases in many models. A growing body of research has shown that SAL has significant beneficial antioxidant effects in many neurological diseases [28].

MDA is a major aldehyde product of lipid peroxidation. Enzymatic antioxidants include SOD, CAT, and GSH-Px [45]. Recent studies have shown increased MDA and decreased SOD, CAT, and GSH levels in kidney animals with diabetes than those in control kidneys. Our previous study showed that salidroside could inhibit the ferroptosis of neurons in AD mice [46]. In this study, results showed that MDA was elevated and SOD and GSH were decreased in SAMP8 group kidneys. However, SAL treatment inhibited MDA and increased SOD and GSH levels in the kidneys of the SAMP8 group. Furthermore, GPX4 levels were lower in the kidneys of the SAMP8 group than in SAMRI mice, whereas GPX4 levels were elevated in the kidneys of SAL-treated groups. Although GPX4 disruption is repaired by SAL, upregulation of SLC7A11 expression has been reported to cure the disease by inhibiting ferroptosis. SLC7A11 levels decreased in the kidney tissues of the SAMP8 group. However, it was restored by SAL.

This study provides evidence that SAL exerts a protective effect against aging-related kidney disease by inhibiting ferroptosis. Therefore, this study provides a strategy for the treatment of ferroptosis-related diseases that are caused by aging and lays the foundation for the development of new therapeutic drugs for ferroptosis-related diseases. However, this study explored only the relationship between SAL, ferroptosis, and renal interstitial fibrosis in the context of aging, and there is no related research conducted on the mechanisms involved, which need to be further explored in the future.

## 4. Materials and Methods

### 4.1. Screen Targets of SAL and Aging-Related Renal Fibrosis

We hypothesize that the targets of SAL that intersect with the targets of aging-related renal fibrosis were potential therapeutic targets of SAL in SAMP8 mice. The potential targets of SAL were retrieved from ChemMapper databases (http://www.lilab-ecust.cn/chemmapper/index.html) (search date: 13 July 2022), PharmMapper databases (http://lilab.ecust.edu.cn/pharmmapper/index.php) (search date: 13 July 2022), Similarity ensemble approach (SEA, http://sea.bkslab.org/) (search date: 14 July 2022), SuperPred databases (http://prediction.charite.de/) (search date: 14 July 2022), and SwissTargetPrediction (http://www.swisstargetprediction.ch/) (search date: 14 July 2022). The therapeutic targets for aging-related renal fibrosis were obtained from the GeneCards databases (https://previous.genecards.org/) (search date: 8 July 2022) and phenolyzer databases (https://phenolyzer.wglab.org/) (search date: 8 July 2022).

### 4.2. Enrichment Analysis and Network Construction

The enrichment analysis was performed by the SangerBox (http://sangerbox.com (accessed on 1 October 2022)) and cluster profile (http://yixuetongji.top/tools.html (accessed on 1 October 2022)), which can classify molecular function, biological process, cellular components, and the KEGG pathway. The screened common targets between SAL and aging-related renal fibrosis were analyzed to construct a PPI network. The protein subnetworks were constructed by the MCODE algorithm (http://baderlab.org/Software/MCODE (accessed on 1 October 2022)).

### 4.3. Experimental Design

Three-month-old male SAMP8 mice and SAMR1 mice were obtained from the Department of Medicine (Department of Experimental Animal Science) of Peking University and were fed in the SPF-level experimental animal center of Southern Medical University. The experiment was approved by the Animal Ethics Committee of Southern Medical University (Approval number: L2019185). Groups were designed when mice were 5 months, including the SAMP8+saline (i.g., n = 10), SAMP8 + 30 mg/kg/day salidroside (Macklin, lot: C10739039) (i.g., n = 8), SAMP8 + 60 mg/kg/day salidroside (i.g., n = 8), and SAMP8 + 5 mg/kg/day Ferrostatin-1 (Fer-1, Topscience, lot: 347174-05-4) (i.p., n = 8) groups. Fer-1 is a potent inhibitor of ferroptosis and was used as a positive control [47]. SAMR1 mice were used as the control group (saline, i.g., n = 10). All the mice were treated for 3 months. After the intraperitoneal injection of pentobarbital sodium, blood was collected from the heart. The abdominal cavity was opened and the kidney removed. One kidney tissue was fixed in formalin and embedded in paraffin for pathological staining, while the rest was quickly frozen in liquid nitrogen and preserved at −80 °C for further study.

### 4.4. Biochemical Analysis

Renal function was assessed by measuring albumin (ALB), blood urea nitrogen (BUN), and serum creatinine (SCr) in mice. Aortic blood obtained from anesthetized mice was used to measure SCr (C011-2-1JianChen Nanjing), BUN (C013-1-1JianChen Nanjing), and ALB (A028-2-1JianChen Nanjing) levels.

### 4.5. Histology and Morphometry

Kidneys were removed and fixed with 4% paraformaldehyde for 24 h at 4 °C. Sections (5 μm) were cut from paraffin-embedded kidney tissues. Sections were stained with hematoxylin–eosin (HE) and Masson’s trichrome staining for h analysis. Masson staining was used to quantify fibrosis. Sections were made to detect collagen fibers. The area of interstitial fibrosis was identified, excluding the vascular area of the area of interest, such as interstitial fibrosis or interstitial fibrosis. Collagen was deposited to the total tissue area. Tubular. ImageJ software (http://rsb.info.nih.gov/ij, access date: 17 April 2022) was used. In addition, a qualitative assessment was included, which was performed blindly by a renal pathologist. The sections for immunohistochemical staining were incubated with a primary antibody against GPX4 overnight at 4 °C, and secondary anti-goat IgG (Proteintech, SA00001-2, Chicago, IL, USA) for 30 min at room temperature. DAB color development, hematoxylin counter-staining, and dehydrated transparent mount section were obtained. The sections were stored in the dark and eventually photographed with digital slice scanner (KF-PRO-005, KFBIO, China).

### 4.6. Kidney Iron Staining

The paraffin sections (5 µm) were dewaxed, hydrated, and then immersed in TBST containing 3% H_2_O_2_ for 10 min. The sections were then treated with an equal-ratio mixture of 4% aqueous potassium ferrocyanide and 4% hydrochloric acid for 30 min. Iron staining was amplified with TBS containing 0.025% DAB and 0.0033% H_2_O_2_ for 10 min. All sections were handled simultaneously to maintain consistency in the dyeing conditions. The sections were stained with hematoxylin, differentiated, sealed, and observed.

### 4.7. Measurement of Malondialdehyde (MDA), Superoxide Dismutase (SOD), and Glutathione (GSH)

The MDA, SOD, and GSH levels in tissues were detected using a lipid peroxidation MDA assay kit, GSH assay kit, and SOD assay kit (Beyotime, Shanghai, China), respectively, following the instructions of the manufacturer.

### 4.8. Polymerase Chain Reaction

Total RNA was extracted from the kidney tissues using TRIzol reagent (Invitrogen) and converted to cDNA using the HiScript ^®^ III-RT SuperMix for qPCR (+gDNA wiper) kit (Vazyme Biotech Co., Ltd.); qPCR was performed using ChamQ Universal SYBR qPCR Master Mix (Vazyme Biotech Co., Ltd.) in Roche LightCycler96. The relative mRNA levels were calculated by normalization to the GAPDH levels (B661304, Sangon Biotech, Shanghai, China). Relative gene expression was analyzed based on fold-change (2^− ΔΔ Ct^ method). Primer sequences were shown in Table 2.

### 4.9. Western Blotting

The kidney tissues of mice in each group were added with protein lysate, and the total protein was extracted after homogenization. The protein concentration of each group was calculated by the BCA method and denatured at 5 × loading buffer at 95 °C for 10 min. The total protein loading of each group was 20 μg. Protein samples were electrophoresed in SDS–polyacrylamide gel electrophoresis (SDS-PAGE) and transferred onto PVDF membranes (Millipore, Billerica, MA, USA). The blots were blocked with 5% skimmed milk. Then, they were probed with primary antibodies: GPX4 (ab125066, 1:2000, Abcam, Cambridge Science Park, Cambridge, UK), SLC7A11 (26864-1-AP, 1:1000, Proteintech, Chicago, IL, USA), ACSL4 (ab155282, 1:10,000, Abcam, Cambridge Science Park, Cambridge, UK), TFR1 (bs-21319R, 1:1000, Bioss, Bejing, China), FTH1 (#3998, 1:1000, CST, Boston, MA, USA), FPN1 (2660, 1-1-AP, Proteintech, Chicago, IL, USA), α-sma (# 19245, 1:2000, CST, Boston, MA, USA), and TGF-β1 (#84912, 1:1000, CST, Boston, MA, USA) overnight at 4 °C. The blots were washed and incubated for 1 h at room temperature with the HRP-conjugated secondary antibody and then developed with enhanced chemiluminescence reagents (E412-01, Vazyme, Nangjing, Jiangsu, China). The densitometry of the protein bands was quantified using ImageJ software.

### 4.10. Molecular Docking

The main compounds of SAL and aging-related renal fibrosis protein targets were analyzed by molecular docking using the AutoDock Vina and AutoDock. The 3D structures of key protein targets were obtained from the RCSB Protein Data Bank (PDB) and AlphFold databases. The figures of the active binding site were generated with the PyMOL software.

### 4.11. Statistical Analysis

Quantitative data are represented as mean ± Standard Deviation (SD). Statistical analyses were completed using SPSS version 25. Comparison among groups was analyzed using a one-way analysis of variance (ANOVA) test with Tukey post hoc multiple comparisons. Results were considered statistically significant when *p* < 0.05. Data were tabulated and plotted using GraphPad Prism, version 8.

## 5. Conclusions

After the intervention of SAL, renal fibrosis and ferroptosis in 8-month-old SAMP8 mice can be alleviated, which may be related to regulating renal iron metabolism, reducing iron deposition, regulating SLC7A11 and GPX4 protein expression, and finally alleviating renal ferroptosis. SAL delays renal aging and inhibits aging-related glomerular fibrosis by inhibiting ferroptosis in SAMP8 mice.

## Figures and Tables

**Figure 1 molecules-27-08039-f001:**
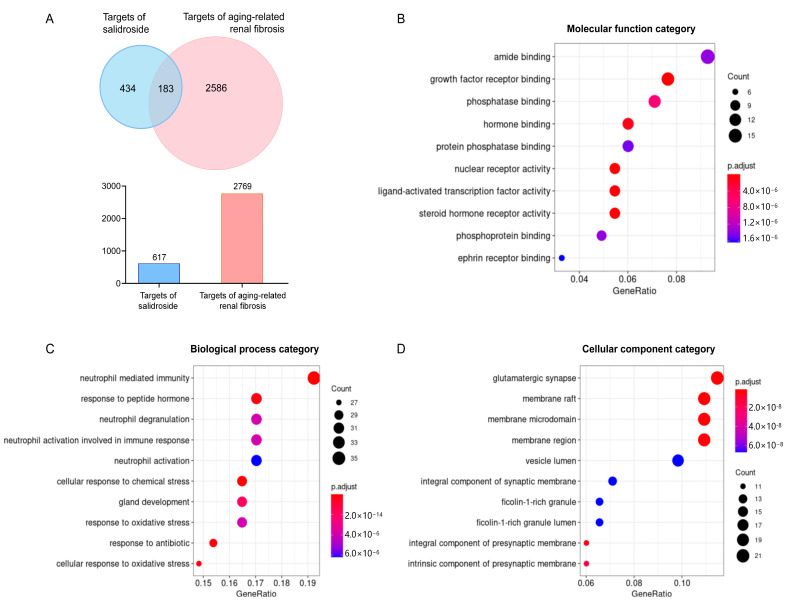
Potential-target screening and enrichment analysis. (**A**) The predicted targets of SAL and aging-related renal fibrosis. A total of 183 overlap targets were screened by the Venn diagram. The bubble maps of molecular function (**B**), biological process (**C**), and cellular component (**D**).

**Figure 2 molecules-27-08039-f002:**
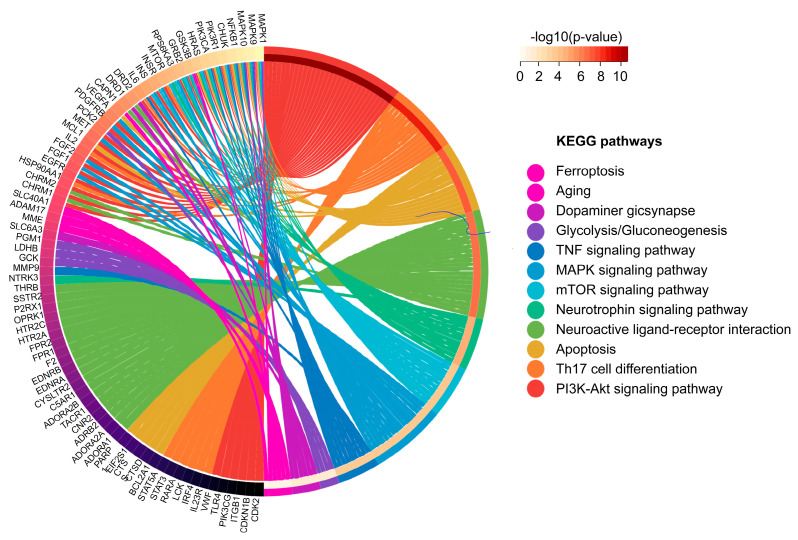
Circle diagram of the significant KEGG pathways.

**Figure 3 molecules-27-08039-f003:**
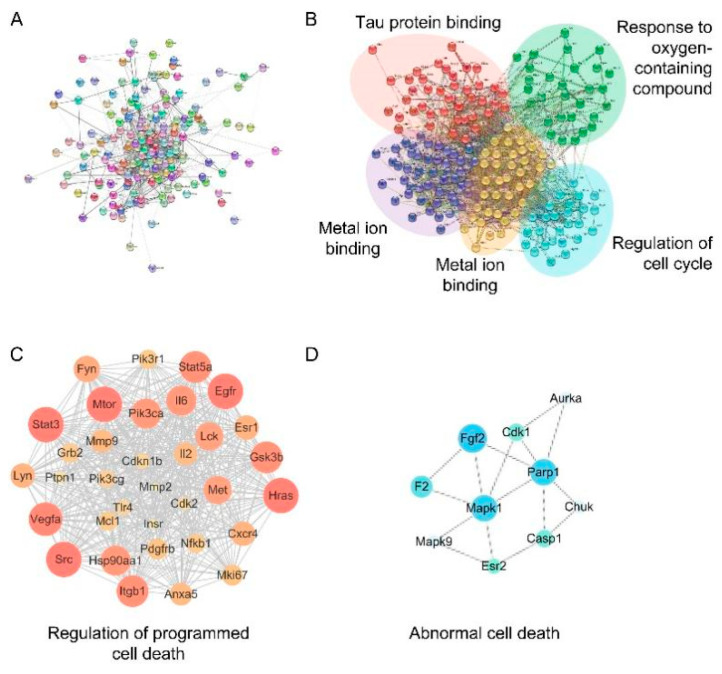
Network cluster analysis. (**A**) PPI network of 183 potential therapeutic targets. (**B**) Network cluster analysis. The main function of each cluster was identified by enrichment analysis. (**C**,**D**) High-scoring subnetworks and their major function.

**Figure 4 molecules-27-08039-f004:**
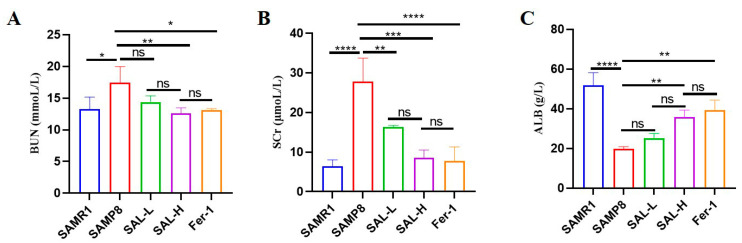
Effects of SAL on renal function parameters in the serum of SAMP8 mice. (**A**) The blood urea nitrogen (BUN) serum level. (**B**) The serum creatinine (SCr) level. (**C**) The albumin (ALB) serum level. Data are expressed as means ± SD, * *p* < 0.05, ** *p* < 0.01, *** *p* < 0.001, **** *p* < 0.0001.

**Figure 5 molecules-27-08039-f005:**
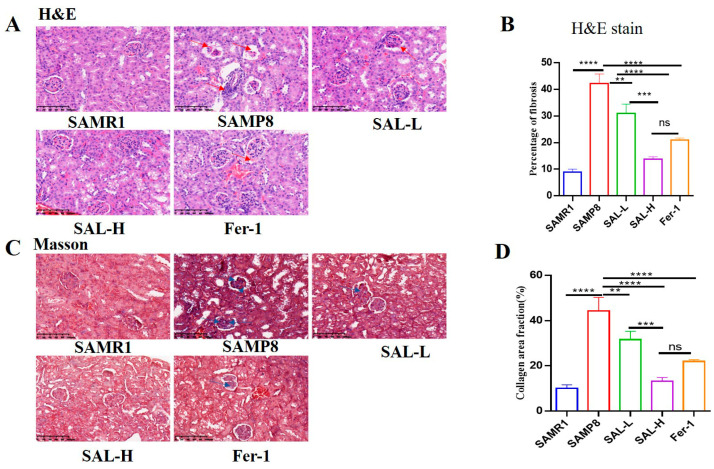
Effects of SAL on histopathological changes in the renal cortex in SAMP8 mice: hematoxylin–eosin (H&E) and Masson staining, 400×. (**A**,**B**) The results of HE staining in the renal cortex. The arrows indicate the glomerular mesangial cells. (**C**,**D**) The results of Masson staining in the renal cortex. The arrows indicate glycoprotein deposition. Data are expressed as means ± SD, ** *p* < 0.01, *** *p* < 0.001, **** *p* < 0.0001.

**Figure 6 molecules-27-08039-f006:**
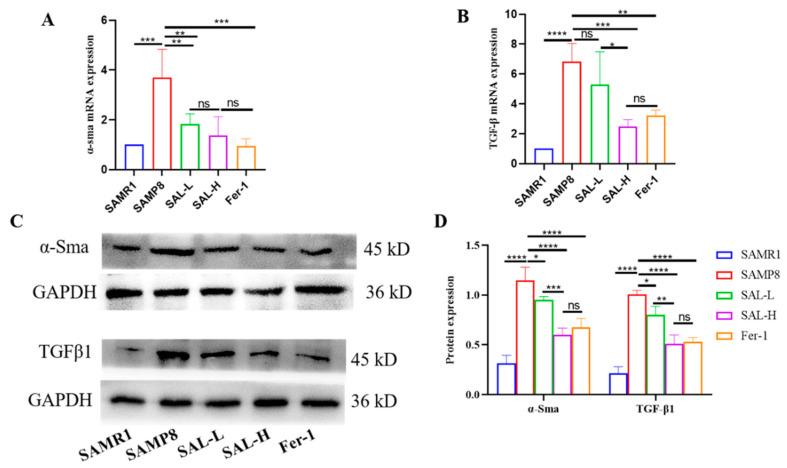
Effects of SAL on expressions of α-SMA and TGF-β1 in renal tissue of SAMP8 mice (Western blot and q-PCR). (**A**) The relative expression of α-SMA. (**B**) The relative expression of TGF-β1. (**C**,**D**) Western blot analysis of α-SMA and TGF-β1 expression, and quantification in the kidney. Data are expressed as means ± SD, * *p* < 0.05, ** *p* < 0.01, *** *p* < 0.001, **** *p* < 0.0001.

**Figure 7 molecules-27-08039-f007:**
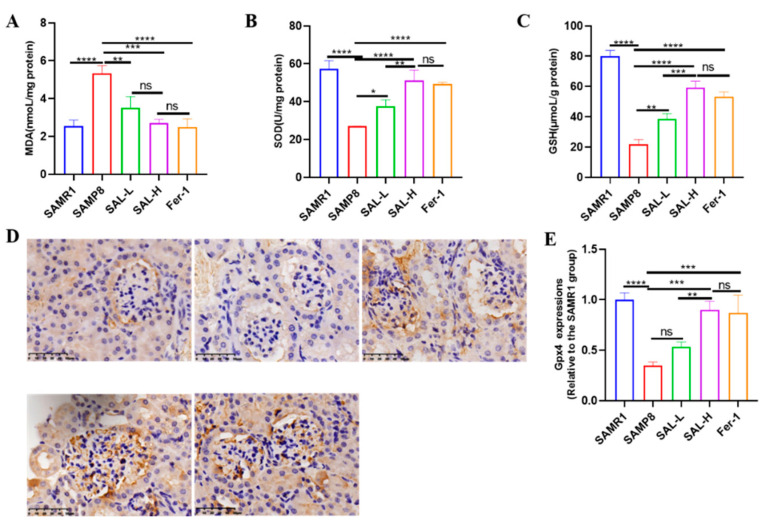
Effects of SAL on the lipid peroxidation and GPX4 levels of SAMP8 mice. (**A**) The MDA levels. (**B**) The SOD levels. (**C**) The GSH levels. (**D**,**E**) The GPX4 levels (IHC). Data are expressed as means ± SD, * *p* < 0.05, ** *p* < 0.01, *** *p* < 0.001, **** *p* < 0.0001.

**Figure 8 molecules-27-08039-f008:**
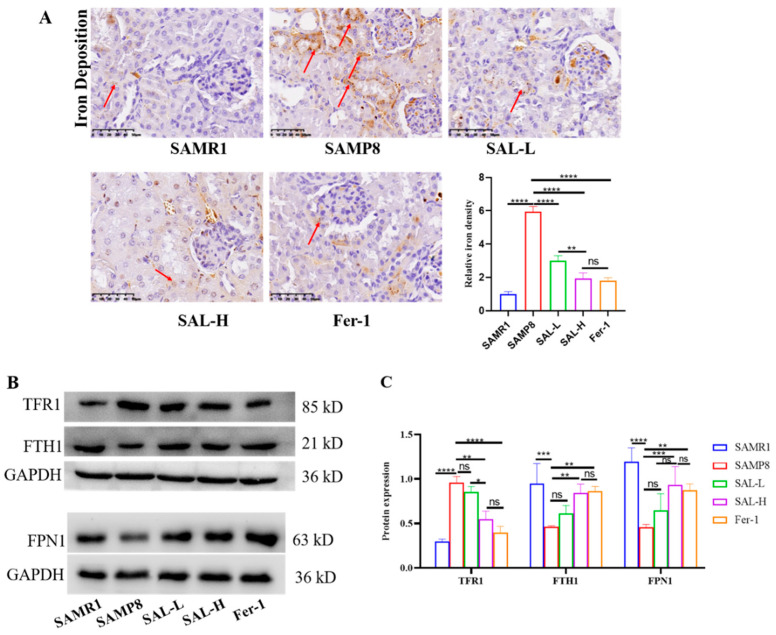
The effects of SAL on kidney iron deposition and the expression of iron transport-related proteins in SAMP8 mice. (**A**) Kidney iron deposition magnification × 400. (**B**,**C**) The expression of FTH1, TFR1, and FPN1 proteins in the kidney of SAMP8 mice. Data are expressed as means ± SD, * *p* < 0.05, ** *p* < 0.01, *** *p* < 0.001, **** *p* < 0.0001.

**Figure 9 molecules-27-08039-f009:**
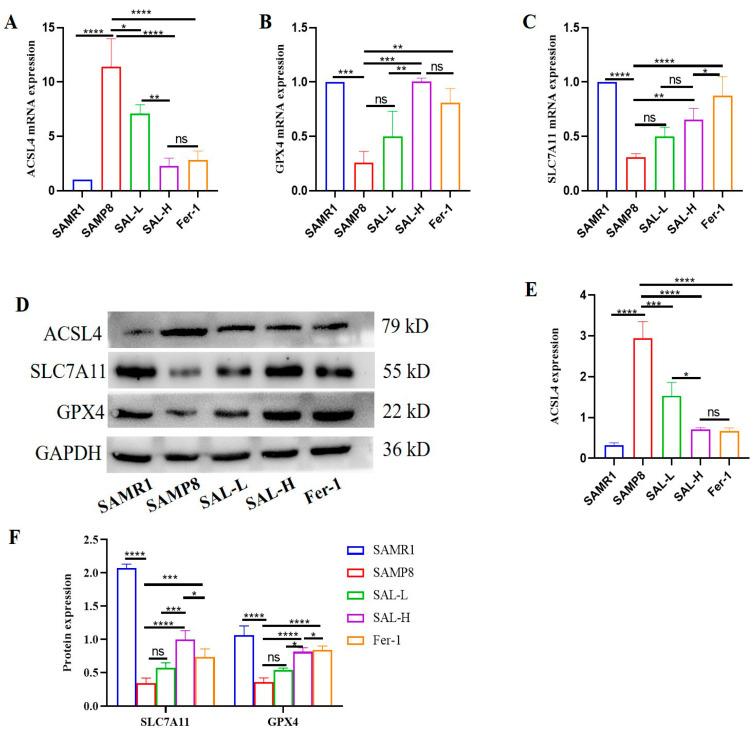
Effects of SAL on the expressions of ACSL4, SLC7A11, and GPX4 in renal tissue of SAMP8 mice (Western blot and q-PCR). (**A**) The relative expression of ACSL4. (**B**) The relative expression of SLC7A11. (**C**) The relative expression of GPX4. (**D**–**F**) Western blot analysis of ACSL4, SLC7A11, and GPX4 expression, and quantification in the kidney. Data are expressed as means ± SD, * *p* < 0.05, ** *p* < 0.01, *** *p* < 0.001, **** *p* < 0.0001.

**Figure 10 molecules-27-08039-f010:**
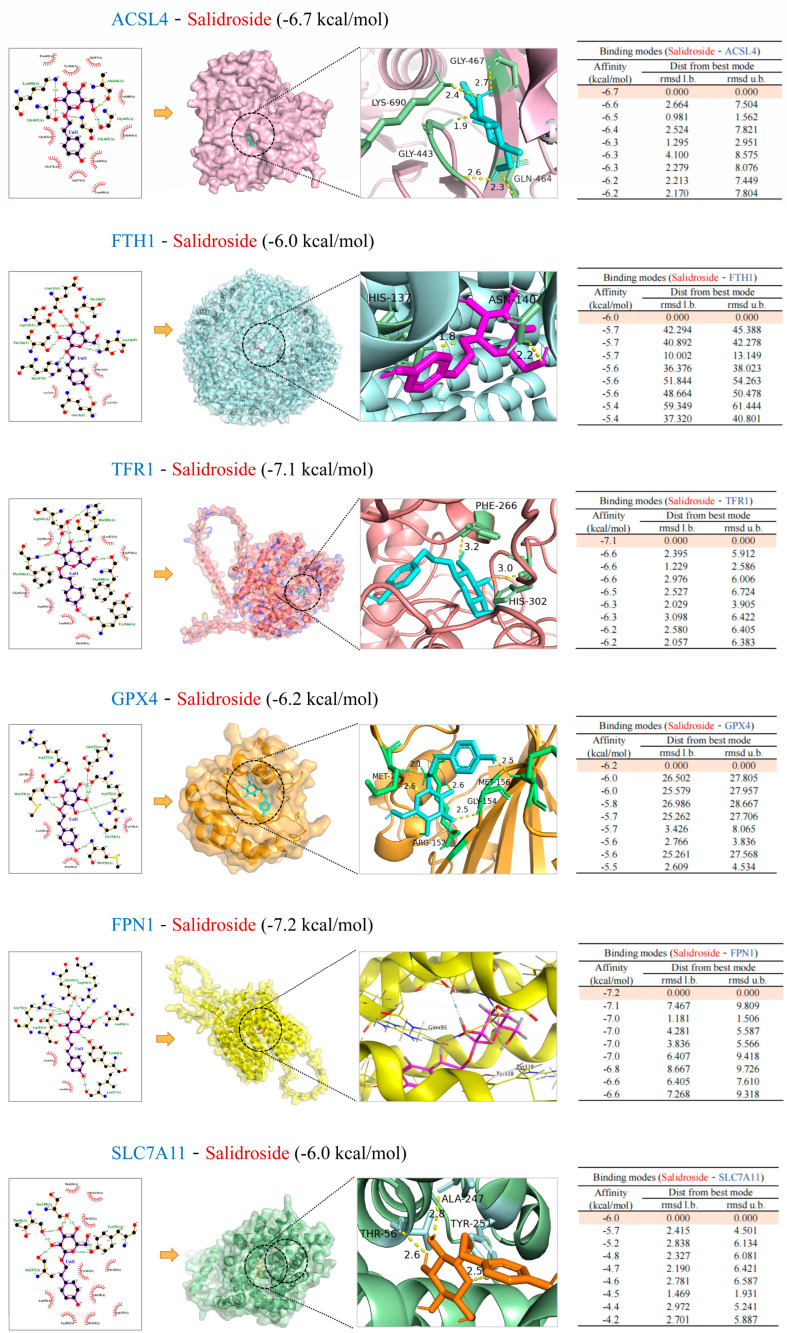
Diagram of molecular docking patterns between SAL and vital ferroptosis protein targets. The docking positions of the binding mode with the best binding energy are shown from global and local perspectives, respectively. The 2D binding modes and detailed docking solutions are presented.

**Table 1 molecules-27-08039-t001:** Potential targets ID.

Uniprot ID	PDB ID	AlphaFord ID	Gene Symbol
P36970	5l71		GPX4
D4ADU2		Q9WTR6	SLC7A11
O35547		Q9QUJ7	ACSL4
P19132	5xb1		FTH1
Q99376		Q62351	TFR1
Q923U9		Q9JHI9	FPN1

**Table 2 molecules-27-08039-t002:** Gene primer sequence used for qRT-PCR.

GPX4	Forward primer	CCGCCGAGATGAGCTGG
	Reverse primer	GTCGATGTCCTTGGCTGAG
SLC7A11	Forward primer	ATGGTCAGAAAGCCAGTTGTG
	Reverse primer	CAGGGCGTATTACGAGCAGT
ACSL4	Forward primer	TCCCTGGACTAGGACCGAAG
	Reverse primer	GGGGCGTCATAGCCTTTCTT
TGF-β	Forward primer	CCAGATCCTGTCCAAACTAAGG
	Reverse primer	CTCTTTAGCATAGTAGTCCGCT
α-sma	Forward primer	CTATGAGGGCTATGCCTTGCC
	Reverse primer	GCTCAGCAGTAGTAACGAAGGA
β-actin	Forward primer	CCACCATGTACCCAGGCATT
	Reverse primer	CGGACTCATCGTACTCCTGC

## Data Availability

Data are contained within the article and Appendix A.

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
