# Peer review of "Salidroside Alleviates Renal Fibrosis in SAMP8 Mice by Inhibiting Ferroptosis"

_molecules, 2022, doi:10.3390/molecules27228039_

Round 1

Reviewer 1 Report

1.   Figure 4-9: Figures show “****”, but the note on the figure lacks description.

2.   Figure 5. Arrows indicating the glomerular mesangial cells and glycoprotein deposition were not seen in HE and Masson staining. HE staining results were not scored.

3.   Page 5/16, line 130: Figure C – Figure 5C.

4.   Figure 6C, Figure 8B. The shape of the protein band is different from that of the GAPDH. Please check the original data.

5.   Figure 7E lack illustration.

6.   Figures 8 to 10 described in the results do not match the serial numbers of the Figures.

7.   PDB IDs of GPX4, SLC7A11, ACSL4, FTH1, TFR1, and FPN1 were not labeled for molecular docking.

8.   There is a lack of discussion on the relationship between the research contents of network pharmacology and the experimental research contents.

9.   Conclusion is too simple.

Author Response

Dear Editorial Board of Molecules,

Thank you for your kind reply to our manuscript entitled “Salidroside alleviates renal fibrosis in SAMP8 mice by inhibiting Ferroptosis”(ID:molecules-1948667). Those comments are all valuable and very helpful for revising and improving our paper, as well as the important guiding significance to our researches. We have studied comments carefully and have made correction which we hope meet with approval. Revised portion are marked in red in the paper. The main corrections in the paper and the responds to the reviewer’s comments are as flowing:

Responds to the reviewer’s comments:

Reviewer #1:

1.Figure 4-9: Figures show “****”, but the note on the figure lacks description.

Response: Thank you very much for your comments. We have added the description to Figures 4-9.

  1. Figure 5. Arrows indicating the glomerular mesangial cells and glycoprotein deposition were not seen in HE and Masson staining. HE staining results were not scored.

Response: Thank you very much for your comments. HE and Masson staining for glomerular Mesangial cells and glycoprotein deposition arrows have been added in Figure 5. The statistical chart of HE staining results has been supplemented.

  1. Page 5/16, line 130: Figure C – Figure 5C

Response: Thank you very much for your comments. Corrected “Figure C”to “Figure 5C”.

  1. Figure 6C, Figure 8B. The shape of the protein band is different from that of the GAPDH. Please check the original data.

Response: Thank you very much for your comments. We have checked the band and the it has been provided with the manuscript at the beginning of the submission system.

  1. Figure 7E lack illustration.

Response: Thank you very much for your comments. Related illustration has been added, the contents are as follows: Furthermore, we measured GPX4 expression in the mice kidneys. Immunohistochemistry showed that GPX4 expression decreased in the kidneys of the SAMP8 group; however, for the SAL and Ferrostatin-1 groups, there was an increase in the GPX4 ex-pression (Figure 7D-E).

  1. Figures 8 to 10 described in the results do not match the serial numbers of the Figures

Response: Thank you very much for your comments. This is our negligence and the relevant serial number has been changed.

  1. PDB IDs of GPX4, SLC7A11, ACSL4, FTH1, TFR1, and FPN1 were not labeled for molecular docking.

Response: Thank you very much for your comments. We have added the details of these proteins used in molecular docking were listed in Table 1.

8.There is a lack of discussion on the relationship between the research contents of network pharmacology and the experimental research contents.

Response: Thank you very much for your comments. We have added the discussion of the relationship between network pharmacology and the experimental research in “Discussion”, on page 12/18 , lines 272-282.

With the progress of high-throughput technologies, biomedical data has greatly accumulated and continues to grow rapidly. Network pharmacology provides a feasible way to obtain an overall understanding of traditional Chinese medicine (TCM) prescriptions from these massive clinical and experimental data. Therefore, the combination of network pharmacology and experimental research is a promising approach for identifying potential targets and uncovering therapeutic mechanisms. In this study, 183 potential therapeutic targets of salidroside were screened by network pharmacology for further research. Enrichment analysis and PPI network analysis found that salidroside can regulate metal ion binding, tau protein binding, and oxygen-containing compounds via the ferroptosis pathway. Based on the network pharmacology findings, the following experimental research was focused on ferroptosis.

  1. Conclusion is too simple.

Response: Thank you very much for your comments. We have revised the conclusion on page 16/18, lines 433-436. the contents are as follows: After the intervention of SAL, renal fibrosis and ferroptosis in 8-month-old SAMP8 mice can be alleviated, which may be related to regulating renal iron metabolism, reducing iron deposition, regulating SLC7A11 and GPX4 protein expression, and finally alleviating renal ferroptosis. SAL delays renal aging and inhibits aging-related glomerular fibrosis by inhibiting ferroptosis in SAMP8 mice.

We tried our best to improve the manuscript and made some changes in the manuscript. These changes will not influence the content and framework of the paper. And here we did not list the changes but marked in red in revised paper. If the reviewers still have comments, we sincerely hope that you can bring them up for discussion

We appreciate for Editors/Reviewers’ warm work earnestly, and hope that the correction will meet with approval.

Once again, thank you very much for your comments and suggestions.

Reviewer 2 Report

The manuscript Salidroside alleviates renal fibrosis in SAMP8 mice by inhibiting Ferroptosis by Sixia Yang et al. provides evidence that salidroside improves renal function and reduces fibrosis in the senescence-accelerated mice SAMP8. The authors associate salidroside effects with ferroptosis and demonstrate that salidroside treatment reduces oxidative stress, iron overload, and TGFbeta-expression. 

The authors´ study focuses on aging-related renal fibrosis. However, in the introduction, methodology, and discussion, they barely address how fibrosis develops in their model. Moreover, several studies have uncovered molecular mechanisms in SAMP8 mice kidneys. How does the current literature on aging-related fibrosis relate to their findings? Can their findings be translated to other types of fibrosis? What are the limitations of their findings using this model? 

Materials and methods. The authors do not mention the sex of the mice included in their study nor the study´s approval number. How did they sacrifice the mice? What was their anesthetic protocol? 

In section 4.5. the authors only describe histology, not morphometry. For example, how many fields per image were included? How were they analyzed/quantified? Also, there is no information regarding the microscope used to obtain the images.

Section 4.7 PCR: information about the thermal cycler used is missing

Section 4.7 Western blot: information about protein lysates and how they were obtained is missing.

For the in silico methodology (screening and docking analyses), the authors should include the URLs of the different platforms used and, when applicable, the dates of the searches. Also, the raw data summarized in Figure 1A (434 targets of salidroside, 2586 targets of age-related fibrosis, and 183 targets intersecting in the Venn diagram) should be included as a supplementary file. 

Figure 5 C mentions the collagen volume fraction. Volume is incorrect, it should be area.

Figures 7 D and E are not described adequately in the figure legend. 

The reference section is missing. Also, the format needs to follow the journal guidelines. Same for the order of the manuscript´s sections. 

Line 37: asecretion -> secretion

Line 42: Recent study -> A recent study

Line 46: oxidation -> oxidant

Line 62: remove including

Line 64: UUO -> define UUO abbreviation

Line 65: the -> this

Line 85 and Figure 2: dopaminer gicsynapse -> dopaminergic synapse

Author Response

Dear Editorial Board of Molecules,

Thank you for your kind reply to our manuscript entitled “Salidroside alleviates renal fibrosis in SAMP8 mice by inhibiting Ferroptosis”(ID:molecules-1948667). Those comments are all valuable and very helpful for revising and improving our paper, as well as the important guiding significance to our researches. We have studied comments carefully and have made correction which we hope meet with approval. Revised portion are marked in red in the paper. The main corrections in the paper and the responds to the reviewer’s comments are as flowing:

Responds to the reviewer’s comments:

Reviewer 2:

1.The authors´ study focuses on aging-related renal fibrosis. However, in the introduction, methodology, and discussion, they barely address how fibrosis develops in their model. Moreover, several studies have uncovered molecular mechanisms in SAMP8 mice kidneys. How does the current literature on aging-related fibrosis relate to their findings? Can their findings be translated to other types of fibrosis? What are the limitations of their findings using this model?

Response: Thank you very much for your comments. We have added the relevant content to the introduction section, on page 2/18, lines 40-63, as follows:The senescence-accelerated mouse(SAM) strains consist of senescence-prone inbred strains (SAMP) and senescence-resistant inbred strains (SAMR). SAMP8 mice is one of the senescence-accelerated prone mice, based on the grading score data of aging, life span, and pathological phenotype, Kyoto University conducted selective in-breeding on AKR/J mouse strain donated by Jackson Laboratory in 1968, which is characterized by learning and memory deficits and impaired immune function. The serum oxidative stress level of SAMP8 mice was higher than that of SAMR1 mice, and the expression levels of muscle atrophy, inflammation fibrosis-related, and aging gene marker genes in the elderly SAMP8 mice were significantly different from those in the young control group. Elderly SAMP8 mice can show adverse cardiac remodeling, liver fibrosis, and renal fibrosis, which are age-related pathologies, and their incidence rate and severity increase with age.

Studies have shown that age-related mitochondrial dysfunction is closely related to renal fibrosis. There is dysfunction in the mitochondria of elderly SAMP8 mice. Renal pathological changes of 9-month-old SAMP8 mice include renal tubulointerstitial fibrosis and focal segmental glomerulosclerosis. Renal fibrosis of SAMP8 mice is related to age. The Wnt/β-catenin/RAS signaling pathway was activated in the kidney of SAMP8 mice. Compared with the aging SAMR1 control group, the consumption lev-el of GSH/GSSG ratio and MDA level in the 6-month-old-month old SAMP8 mice were increased. Early CKD was observed in aged SAMP8 mice. The spontaneous model such as SAM has obvious advantages over the gene-modified model, which can better reflect the changes in age-related diseases. It is hoped that it can be used more widely for the research of biological gerontology resources. There was ferroptosis and in-creased lipid peroxidation levels in the muscle of SAMP8 mice.

2.Materials and methods. The authors do not mention the sex of the mice included in their study nor the study´s approval number. How did they sacrifice the mice? What was their anesthetic protocol?

Response: Thank you very much for your comments. This is our negligence in writing the article, and the relevant content has been added to the material and method section, on page14/18 , lines 352-365. the experiment approval number has been added. The details are as follows:After the intraperitoneal injection of pentobarbital sodium, blood was collected from the heart. Open the abdominal cavity and remove the kidney. one kidney tissue was fixed in formalin and embedded in paraffin for Pathology related staining, while the rest was quickly frozen in liquid nitrogen. And preserved at-80 °C for further study.

  1. In section 4.5. the authors only describe histology, not morphometry. For example, how many fields per image were included? How were they analyzed/quantified? Also, there is no information regarding the microscope used to obtain the images.

Response: Thank you very much for your comments. The relevant content has been added to the material and method section, page14/18,line 374-384. The details are as follows:Histological analysis. Masson staining was used to quantify fibrosis. Sections were made to detect collagen fibers. Identify the area of interstitial fibrosis, excluding the vascular area of the area of interest, such as interstitial fibrosis or interstitial fibrosis. Collagen was deposited to the total tissue area. Tubular. ImageJ software (http://rsb.info.nih.gov/ij). In addition, a qualitative assessment is included, which is performed blindly by a renal pathologist. The sections for immunohistochemical staining were incubated with a primary antibody against GPX4 overnight at 4 °C, and secondary anti-goat IgG (Proteintech, SA00001-2, USA) for 30 min at room temperature. DAB color development, hematoxylin counter-staining, and dehydrated transparent mount section were obtained. The sections were stored in the dark and eventually photographed with digital slice scanner (KF-PRO-005, KFBIO, China).

  1. PCR: information about the thermal cycler used is missing

Response: Thank you very much for your comments. The relevant content has been added to the material and method section,on page15/18, line 401.

  1. Western blot: information about protein lysates and how they were obtained is missing.

Response: Thank you very much  for your comments. The relevant content has been added to the material and method section, page16/18, line 407-410, The details are as follows:The kidney tissues of mice in each group were added with protein lysate, and the total protein was extracted after homogenization. The protein concentration of each group was calculated by the BCA method, and denatured at 5 × loading buffer,95 ℃ for 10 min. The total protein loading of each group was 20 μg.

  1. For the in silico methodology (screening and docking analyses), the authors should include the URLs of the different platforms used and, when applicable, the dates of the searches. Also, the raw data summarized in Figure 1A (434 targets of salidroside, 2586 targets of age-related fibrosis, and 183 targets intersecting in the Venn diagram) should be included as a supplementary file.

Response: Thank you very much  for your comments. The URLs and search date of the platforms have added on page 14/18 ,line 336-343 . The details of potential targets of salidroside and age-related fibrosis, and 183 intersect targets were listed in Table S1.

  1. Figure 5 C mentions the collagen volume fraction. Volume is incorrect, it should be area.

Response: Thank you very much  for your comments. We have made corresponding changes.

  1. Figures 7 D and E are not described adequately in the figure legend.

Response: Thank you very much  for your comments. We have made corresponding changes, the contents are as follows: Furthermore, we measured GPX4 expression in the mice kidneys. Immunohistochemistry showed that GPX4 expression decreased in the kidneys of the SAMP8 group; however, for the SAL and Fer-1 groups, there was an increase in the GPX4 ex-pression (Figure 7D-E).

  1. The reference section is missing. Also, the format needs to follow the journal guidelines. Same for the order of the manuscript´s sections.

Response: Thank you very much for your comments. We have added the references according to the target format of the journal.

  1. Line 37: asecretion -> secretion, Line 42: Recent study -> A recent study, Line 46: oxidation -> oxidant, Line 62: remove including, Line 64: UUO -> define UUO abbreviation, Line 65: the -> this,Line 85 and Figure 2: dopaminer gicsynapse -> dopaminergic synapse

Response: Thank you very much for your comments. We have completed the above comments and revised them in the manuscript.

We tried our best to improve the manuscript and made some changes in the manuscript. These changes will not influence the content and framework of the paper. And here we did not list the changes but marked in red in revised paper. If the reviewers still have comments, we sincerely hope that you can bring them up for discussion

We appreciate for Editors/Reviewers’ warm work earnestly, and hope that the correction will meet with approval.

Once again, thank you very much for your comments and suggestions.
